

# Optimization of carton recycling site selection using particle swarm optimization algorithm considering residents' recycling willingness

Yulong Xi, Fengming Tao and Schanelle Brooks

Chongqing University, School of Management Science and Real Estate, Chongqing, China

## ABSTRACT

With the development of the express delivery industry, how to increase the recycling rate of waste cartons has become a problem that needs to be solved. Recycling enterprises began to provide the new recycling mode, door-to-door recycling services, to residents with waste cartons. In this article, we constructed a site selection model for a carton recycling site with the aim of maximizing total profits. Considering the residents' recycling willingness and the government subsidy earned through the contribution to carbon emission reduction, this model achieves the task of site selection and unit price fixation for carton recycling. We used the particle swarm optimization (PSO) algorithm to solve the model and compared it with the genetic algorithm (GA) for validity testing. PSO algorithm was also used to carry out sensitivity analysis in this model. The proposed model and the results of the sensitivity analysis can be used for decision-making in recycling enterprises as well as for further research on waste recycling and reverse logistics.

# INTRODUCTION

The rapid development of E-commerce and the express industry, has led to an increase in the production and distribution of packaging cartons (*Ma et al., 2022*). Though recyclable, their participation in recycling is still low resulting in adverse effects on the economy and the environment (*Tang, 2014*). For the economy, the cost to paper mills of buying pulp raw materials to produce new cartons is much higher than collecting waste cartons for remanufacturing (*Shang et al., 2021*). For the environment, carelessly discarded cartons cause environmental damage, and the excessive woodcutting for carton production poses great ecological harm, such as increasing the global carbon dioxide content and leading to more extreme climate (*Su et al., 2020*). On the contrary, recycling cartons can not only save production costs and bring better economic benefits, but also promote carbon emission reduction (*Tong et al., 2023*; *Cho et al., 2022*). China's "Energy Production and Consumption Revolution Strategy (2016–2030)" clearly states that the $CO_2$ emissions per unit of GDP in China are expected to decrease by 18% from 2015 levels by 2020 and decrease by 60–65% from 2005 levels by 2030 (*NDRC & NEA, 2016*). Therefore, it is necessary to increase carton recycling in order to achieve this objective.

Corresponding author
Fengming Tao,
taofengming@cqu.edu.cn

Raising the carton recycling rate requires an efficient recycling system. Therefore, in February 2022, the National Development and Reform Commission and other departments issued the "Guidelines on accelerating the construction of urban environmental infrastructure" (*NDRC et al., 2022*), proposing to improve the regional recycling system of renewable waste resources to help achieve the goal of carbon peaking and carbon neutrality. Now, the recycling of waste resources, including carton recycling, is in a transitional period in China. Traditional carton recycling is mainly carried out using urban floating recycling vendors, however the problem of aging and low efficiency exists when using these mediums (*Liu et al., 2015*; *Li, 2002*), and the method for determining the carton recycling unit price lacks transparency (*Wang, Han & Li, 2008*). Today, the maturity of the Internet has paved the way for developments in the carton recycling industry, allowing for the formation of a new recycling mode namely "Internet + waste recycling" (*Huang & Wei, 2013*). In the new recycling mode, third-party recycling enterprises are born, such as "Mr. Cat" and other enterprises. These recycling enterprises set up recycling sites in certain areas of the city and provide recycling services to the residents who have the demand for recycling waste cartons. After residents place orders online, the staff will come to their doors to collect the cartons and pack them for delivery to the paper mills in the suburbs. Compared with the traditional carton recycling mode, this new mode has the advantages of process institutionalization and price transparency (*Wang et al., 2018*). The new recycling mode greatly improves the efficiency of carton recycling, namely, decreases carbon emissions and increases the output value of the recycling industry. However, due to their short history, the third-party recycling enterprises have to deal with problems like the low recycling rate of cartons and insufficient economic benefits, which are mostly likely the results of unsuitable site distribution and pricing strategies of carton recycling. Therefore, it is necessary to conduct further research on the recycling site selection and pricing amongst recycling enterprises.

The problem of recycling site selection is actually a facility location problem in reverse logistics. From different backgrounds and perspectives, various scholars have proposed models and designed a number of solutions for the facility location problem. *Chauhan & Singh (2016)* used hybrid multi-criteria decision making method to select a sustainable location for a healthcare waste disposal facility. *Zhao et al. (2016)* minimized the total cost and total risk, and solved the hazardous waste recycling site selection problem through multi-objective optimization. *Yadav et al. (2018)* proposed the importance of urban waste recycling and transportation, and established an interval value location model for urban waste recycling and transfer stations under uncertain conditions. *Aydemir-Karadag (2018)* proposed a profit-oriented mixed integer mathematical model, which takes into account both environmental impacts and economic benefits, for the site routing problem of hazardous wastes. *Liu et al. (2019)* constructed an optimization model of recycling plant location using genetic algorithm, and conducted an empirical study with Panyu and Nansha districts in Guangzhou as examples. *Toutouh, Rossit & Nesmachnow (2020)* proposed a series of applications for single-objective and multi-objective heuristic algorithms based on PageRank method and two multi-objective evolutionary algorithms to resolve urban garbage dump sites location problem.

In the research on facility location in the reverse logistics sphere, vast amounts of literature studied the recycling and treatment of non-reusable wastes such as medical, construction and hazardous waste. However, as the implementation of new waste recycling mode are in the early stages, few studies have been conducted on recyclable waste resources such as cartons. There are several new characteristics in the new carton recycling model that have not been discussed in the previous studies. First, unlike other models that serves all nodes, the number of nodes in the new carton recycling model depends on the carton price offered by the recycling enterprise to the residents. Specifically, residents' participation in recycling is contingent upon the unit carton recycling price exceeding their psychological expectation. In other words, only when the unit carton recycling price surpasses what residents psychologically anticipate as fair or satisfactory, they demonstrate willingness to engage in the recycling process (*Minli, Kanglai & Jie, 2018*), otherwise, residents would rather to dump the cartons. Moreover, for the purpose of environmental protection, the contribution of recycled cartons to carbon emission reduction should be considered, and these can act as a certain subsidy for enterprises by the government (*Galinato & Yoder, 2010*).

In addition, scholars have adopted different methods to solve the location model. Some scholars used multi-criteria approaches such as Analytic Hierarchy Process (AHP) (*Aragones-Beltran et al., 2010*), comprehensive network analysis and data envelopment analysis (*Khadivi & Ghomi, 2012*) to conduct site selection research. Some scholars also use meta-heuristic algorithms in the site selection research, such as genetic algorithm (*Liu et al., 2019*), simulated annealing algorithm (*Santosa & Kresna, 2015*) and other meta-heuristic algorithms based on evolutionary strategies. In fact, metaheuristic algorithms are designed to address a broad spectrum of challenging optimization problems, including NP-hard problems like site selection problem, without the need for extensive problem-specific adaptations. In the meantime, meta-heuristic algorithms based on swarm intelligence, such as particle swarm optimization (PSO) algorithm also have very good performance in site selection problems (*Ezugwu et al., 2020*; *Pour & Nosraty, 2006*; *Boonmee, Arimura & Asada, 2018*). PSO has the advantages of fewer required parameters, easy particle coding, fast convergence, and strong universality. Its optimization is mainly based on the fitness function, which is very suitable for solving complex optimization problems (*Wang et al., 2022*; *Yang, Tao & Zhong, 2022*; *Huang, Li & Liu, 2018*), and it is an ideal method to solve NP-hard problems such as the carton recycling site selection problem.

Based on the new characteristics of the new carton recycling mode and the advantages of PSO algorithm, this article focuses on site selection in carton recycling enterprises and constructs a single objective model, including the carbon emission reduction subsidies to maximize the total profit, and applies PSO algorithm to solve the problem. The model validity analysis and sensitivity analysis are then conducted based on a case study to discuss the influence of relevant parameters in the recycling process, so as to provide reference and suggestions for the decision-making of recycling enterprises. In a research sense, reasonable recycling site selection and carton pricing for recycling enterprises can fill the gap in the development of reverse logistics in the new waste recycling mode.

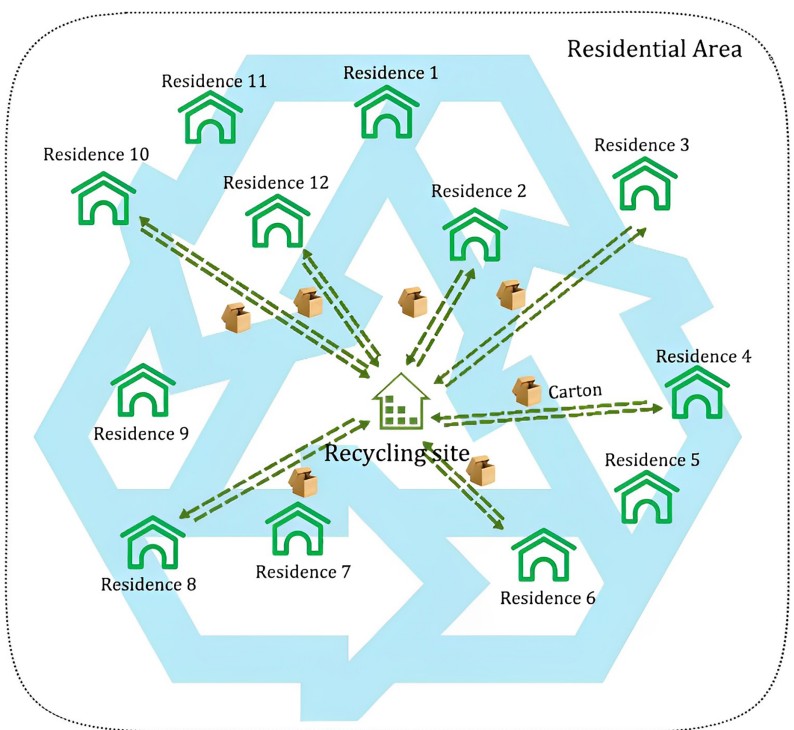

**Figure 1 Diagram of carton recycling.**

## METHODOLOGY

This study is completed from the perspective of a recycling enterprise. The carton recycling site location model includes a single-layer reverse logistics network. See Fig. 1 for the specific diagram. There are two types of nodes in the network. One is a set of resident nodes with certain carton weight, and all their information are known. The other is the recycling site set up by the enterprise for collecting and temporarily storing cartons. Its location and the unit carton recycling price need to be determined.

### Problem description

In the carton recycling process, the enterprise needs to set up a carton recycling site in a certain residential area, and post the unit carton recycling price to residents in the area on the official website. Each resident has a psychological expectation about the unit carton recycling price. Only when the recycling unit price offered by the enterprise is higher than his expectation will he participate in the recycling. The enterprise provides door-to-door pickup service from the site to residence for each resident. After serving all the residents, the enterprise takes all the collected cartons to a suburban paper mill and resells them to the mill at the resale price offered by the mill.

The enterprise's carton recycling cost consists of the cost of recycling cartons from residents and the transportation cost of providing door-to-door services. Since the paper mill is usually far away from the city center, its distance is hardly affected by the location of the recycling site in the urban residential area. Therefore, the transportation cost of sending cartons from the site to the paper mill can be considered as fixed and is not

included in the model. The income of carton recycling comes mainly from the resale of cartons to the paper mill. On this basis, considering the contribution of recycled cartons to carbon emission reduction, the subsidies offered by the government influenced by the total weight of recycling cartons are set as additional income for the enterprise. In the end, the enterprise's profit is equal to its income minus its costs.

In order to help the enterprise to maximize the total profit including the carbon emission reduction subsidies, the location of the carton recycling site and unit carton recycling price considering residents' willingness need to be studied through decision making. In summary, the carton recycling site location model is constructed on the basis of comprehensively considering the interests of residents, the recycling enterprise and their contribution to carbon emission reduction.

## Model assumptions

In order to better reveal the core of the problem and improve the generality of the model proposed in this article, this study is based on the following assumptions.

1. The proposed carton recycling site location model can be used as the basis for future path optimization, but this model does not consider path optimization. The enterprise provides door-to-door pickup service from site to residence.

2. The transportation cost between the site and the paper mill is not considered.

3. The quality of cartons of each resident in the area and his psychological expectation about the unit carton recycling price is known, and cartons are assumed to be identical. Residents who have the recycling willingness will hand over all their cartons. At the same time, due to the characteristics of price transparency of the new recycling mode, the unit carton recycling price offered by the enterprise is uniformly consistent for all residents.

4. For environmental reasons, the enterprise uses electric cars, which consume no fuel and produce no carbon emissions. The transportation cost by distance is known, which is irrelevant to loads. The unit resale price offered by the paper mill is also known, and the carbon emission subsidy based on the total weight of recycling cartons is certain, too.

5. The distance between the site and the resident is assumed to be Euclidean distance, regardless of the fixed cost and capacity limitation of the vehicle for recycling and the influence of the street on the distance.

6. We assume that the location of the site in the region is arbitrary, and only one is selected, regardless of the opening cost of the site.

## Description of parameters

The model parameters are defined as follows.

| | |
|---|---|
| $G$ | Set of all the nodes in the reverse logistics network, $n = \{n\|n = 0, 1, 2, ..., N\}$, 0 is the recycling site. |
| $K$ | Set of all the residents in the reverse logistics network, $i = \{i\|i = 1, 2, ..., N\}$. |
| $p$ | The unit carton recycling price offered by the enterprise (CNY/Kg). |
| $p_i$ | The psychologically expected unit carton recycling price of resident $i$ (CNY/Kg). |
| $p^r$ | The unit carton resale price offered by the paper mill (CNY/Kg). |

| | |
|---|---|
| $d_i$ | The distance from resident $i$ to the recycling site (Km). |
| $w_i$ | The weight of the cartons owned by the resident $i$ (Kg). |
| $c_f$ | The unit transportation price per kilometer (CNY/Km). |
| $\lambda$ | The carbon emission reduction subsidy coefficient (CNY/Kg). |
| $x_i$ | The decision variable, $x_i = 1$ if the residents are willing to participate in carton recycling, otherwise, $x_i = 0$. |

## Mathematical model construction

It can be seen from the problem description that the ultimate goal of the recycling enterprise is to maximize net profit. Therefore, the objective function of the model includes the cost and profit of carton recycling, which is detailed as follows.

(1) The cost of recycling cartons from residents.

This cost is equal to the sum product of the unit carton recycling price offered by the enterprise and the weight of the carton owned by the residents, and only when residents are willing to sell them. It can be expressed mathematically as:

$$Cost_1 = \sum_{i \in K} x_i w_i p \tag{1}$$

(2) The transportation cost.

The transportation cost is equal to the sum product of the round trip distance from the site to the residence and the unit transportation price per kilometer, and only when residents are willing to sell them. It can be expressed mathematically as:

$$Cost_2 = \sum_{i \in K} 2 x_i d_i c_f \tag{2}$$

(3) Income from reselling.

This is the main source of income for the enterprise. Similar to Eq. (1), the income from reselling is equal to the unit carton resale price offered by the paper mill multiplied by the total weight of the collected cartons. It can be expressed mathematically as:

$$Income_1 = \sum_{i \in K} x_i w_i p^r \tag{3}$$

(4) Carbon emission reduction subsidies.

Given the contribution to the reduction in carbon emissions, we assume that the government sets a subsidy for the enterprise to recycle as many cartons as possible. Since the enterprise's main source of income comes from carton resale, profits are significantly affected by fluctuations in unit carton resale price and the enterprise will not be encouraged to increase carton collection if they are not sufficiently subsidized. Therefore, we refer to Juan's literature and setting the subsidy as a coefficient $\lambda$ based on the total weight of recycling cartons (*Juan, Dingyou & Yinggui, 2019*). It can be expressed mathematically as:

$$Income_2 = \sum_{i \in K} x_i w_i \lambda \tag{4}$$

To sum up, the enterprise's final total profit is expressed as:

$$F = Income_1 + Income_2 - Cost_1 - Cost_2 \tag{5}$$

Since the objective function in operations research is generally to find the minimum value, the objective function of the model proposed in this research is expressed as:

$$Min(-F) = \sum_{i \in K} x_i w_i p + \sum_{i \in K} 2 x_i d_i c_f - \sum_{i \in K} x_i w_i p^r - \sum_{i \in K} x_i w_i \lambda \tag{6}$$

Regarding the constraints of the model, we make the following considerations:

(1) The constraint is mainly the price constraint; we require the unit carton recycling price to be a positive number. It can be expressed mathematically as:

$$p \geq 0 \tag{7}$$

(2) Since the enterprise's main income comes from the resale of cartons, the unit carton recycling price must be less than the unit carton resale price. It can be expressed mathematically as:

$$p^r \geq p \tag{8}$$

(3) As mentioned above, residents are willing to participate in carton recycling only when the unit carton resale price offered by the enterprise is higher than their psychological expectation. Therefore, the value of variable $x_i$ is related to the comparison of $p$ and $p_i$ values. It can be expressed mathematically as:

$$x_i = \begin{cases} 1 & p \geq p_i \\ 0 & p \prec p_i \end{cases}, \forall i \in K \tag{9}$$

## ALGORITHM DESIGN
### Characteristics and applications of PSO

PSO is inspired by the study of bird foraging behavior. Its basic idea is to find the optimal solution through the cooperation and information sharing between individuals in the swarm. Each particle separately searches for the optimal solution in the search space, then records and shares the individual optimal value. All particles in a swarm adjust their speed and position based on the current individual optimal value they find and the current global optimal value shared by the whole swarm (*Eberhart & Kennedy, 1995*).

There are many applications of PSO in the logistics field. *Yao, Hu & Yang (2013)* designed the location model of automobile spare part warehouse and solved it with particle swarm optimization algorithm for automobile production and transportation. *Shen & Chen (2017)* proposed a multi-depot model and used PSO algorithm to solve its location and path optimization problems. *Yang, Tao & Zhong (2022)* improved the particle swarm

optimization algorithm by introducing diversity enhancement mechanism and neighborhood search mechanism, and applied it in the optimization of urban wet waste recovery path optimization.

In view of the above carton recycling site location model, combined with previous application experience, we applied PSO algorithm to solve the model.

## PSO principles and calculation procedures

The basic principles of PSO (*Tchomte & Gourgand, 2009*) are as follows.

In the feasible solution space of a D-dimension, there is a particle population P, in which there are *Npop* randomly distributed particles. Each of these particles has their own initial moving speed and initial position, and those in the whole population are optimized iteratively *MaxIt* times. With each iteration, each particle keeps track of two extreme values (*Pbest*, *Gbest*) to update its speed.

Each particle has a velocity and a position, and the initial particle is randomly generated as follows:

$$x_i = rand(1, size). \times (x_{\max} - x_{\min}) + x_{\min} \qquad (10)$$
$$v_i = rand(1, size). \times (v_{\max} - v_{\min}) + v_{\min} \qquad (11)$$

where *size* is the particle length, $[x_{min}, x_{max}]$ is the particle range, and $[v_{min}, v_{max}]$ is the velocity range. Thus, the updated formula of the particle is:

$$v_{ij}(t + 1) = w \cdot v_{ij}(t) + c_1 \cdot rand1_{ij} \cdot (Pbest_{ij}(t) - x_{ij}(t))$$
$$+ c_2 \cdot rand2_{ij} \cdot (Gbest_j(t) - x_{ij}(t)) \qquad (12)$$
$$x_{ij}(t + 1) = x_{ij}(t) + v_{ij}(t + 1) \qquad (13)$$

In the formula, $i = 1, 2, \ldots, N$ is the number of the particle, $X_i = (x_{i1}, x_{i2}, \ldots, x_{iD})$ represents the position of the *i* particle; $V_i = (v_{i1}, v_{i2}, \ldots, v_{iD})$ is the velocity of particle *i*; $pbest_i = (pbest_{i1}, pbest_{i2}, \ldots, pbest_{iD})$ represents the position with the highest fitness value reached by the *i* particle itself (that is, the position of the individual optimal solution in history); $gbest = (gbest_1, gbest_2, \ldots, gbest_D)$ represents the position with the highest fitness value reached by all particles in the population (that is, the position of the global historical optimal solution). Parameter *w* is called the inertia factor, which is used to balance the global and local search ability of particles. $rand1_{ij}$ and $rand2_{ij}$ are two independent uniform random numbers generated within range $[0, 1]$. $c_1$ and $c_2$ are two learning factors that control the influence of social and cognitive components., $t = 1, 2, \ldots$ represents the number of iterations.

*Shi (1998)* introduced a parameter $w_t$, called inertia weight, into the PSO. Since it decreases linearly in the search process, it can affect the velocity and position of each particle, which is used to balance exploration and utilization capabilities and control the convergence behavior of PSO. Inertia weight is added into the algorithm so that the PSO will operate smoothly. Inertial weights are calculated using the following equation:

$$w_t = w_{\max} - \frac{(w_{\max} - w_{\min})}{T} \cdot t \qquad (14)$$

In the formula, $w_t$ decreases linearly as the number of iterations increases, and $w_{\max}$ and $w_{\max}$ represent the maximum and minimum inertial weight values respectively.

Based on the principle of PSO, we now design the specific steps of using PSO in the carton recycling site location model.

Step 1: read the x and y coordinates of residents in the data file, as well as the weight $w_i$ of the cartons owned by the resident $i$ and the psychologically expected unit carton recycling price $p_i$ of carton recycling of Resident $i$.

Step 2: set parameter values in the PSO algorithm, including the maximum number of iterations *MaxIt*, the number of populations *Npop*, the maximum inertia weight $w_{max}$, the minimum inertia weight $w_{min}$, the particle learning factors $c_1$ and $c_2$, the velocity range $(v_{min}, v_{max})$ and the position range (*Varmin, Varmax*).

Step 3: determine the particles encoding. In this research, the x and y coordinate location of the recycling site and the unit carton recycling price are decision variables. Then *Npop* 3D particles are randomly generated as the initial particle swarm, their positions and velocities are generated according to Eqs. (10) and (11). The first two dimensions of the particle represent the x and y coordinates of the site, respectively, and the third dimension represents the unit carton recycling price.

Step 4: calculate the Euclidean distance from the site to each resident, then the fitness of each particle *particle(i).profit* is calculated according to the objective function mentioned in the previous model.

Step 5: set the position of particle $i$ as *pbest$_i$* in the initial particle swarm, and the position of the particle with the largest fitness value as the initial global historical optimal solution *gbest*.

Step 6: the iteration begins, each particle in the population updates its position and velocity according to Eqs. (12)–(14).

Step 7: after the iteration, the global optimal solution in the last generation is output as the result of algorithm calculation.

Figure 2 briefly describes the flow of the algorithm calculation process.

## COMPUTATIONAL EXPERIMENTS

### Algorithm validity test

The Solomon benchmark is a widely recognized dataset that is commonly used to study issues related to vehicle routing problems (VRP). In this section, a Solomon benchmark example set with adjusted parameters (*Bent & Van Hentenryck, 2004*) is used to verify the effectiveness of the PSO. We compare the PSO with the genetic algorithm (GA) by randomly selecting several examples from the original six types. The following experiments were executed 10 times with MatlabR2021a on a PC with Intel core i5 CPU operating at 1.60 GHz. The parameter settings of PSO are presented in Table 1, and the parameter settings of GA are presented in Table 2. The fitness function of these two algorithms is the objective function of the model. In addition, we refer to the price table of

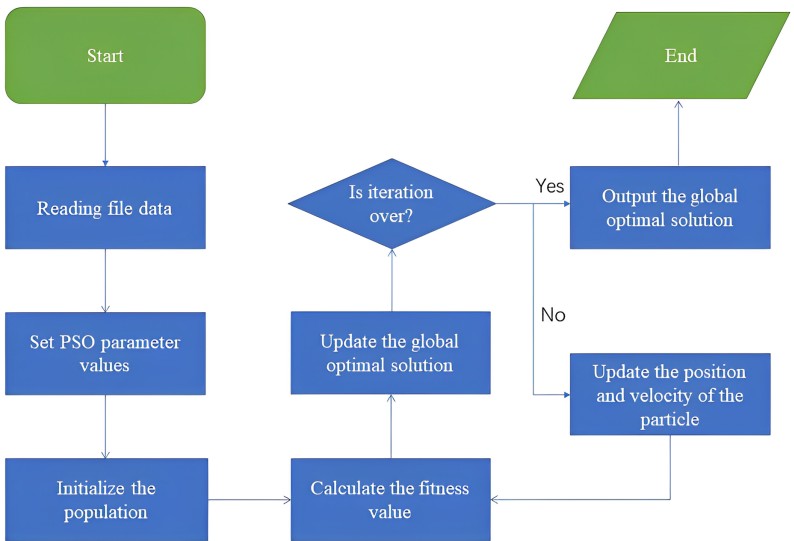

**Figure 2  Flow chart of algorithm calculation process.**

**Table 1  The basic set of parameters of PSO.**

| Parameter | Description | Value |
|---|---|---|
| $Npop$ | Population number | 20 |
| $w_{max}$ | Maximum inertia weight | 0.9 |
| $w_{min}$ | Minimum inertia weight | 0.01 |
| $c_1$ | Personal learning factor | 1.49618 |
| $c_2$ | Global learning factor | 1.49618 |
| $(v_{min}, v_{max})$ | Velocity range | [−10, 10] |
| $(Varmin, Varmax)$ | Position range | [0, 100] |
| $MaxIt$ | Evolution terminate generation | 500 |

**Table 2  The basic set of parameters of GA.**

| Parameter | Description | Value |
|---|---|---|
| $Npop$ | Population number | 20 |
| $p_c$ | Chromosome crossover probability | 0.6 |
| $p_m$ | Chromosome mutation probability | 0.1 |
| / | Chromosome selection method | Tournament |
| $MaxIt$ | Evolution terminate generation | 500 |

domestic waste paper recycling on December 14, 2022, and assume that the unit price of carton resale in the model is 2 CNY/Kg, and refer to the relevant parameters of several new energy waste recycling vehicles released by the Ministry of Industry and Information Technology (*MIIT, 2022*), assume that the energy consumption of vehicles per kilometer is 0.30 KWH, and multiply by the domestic electricity price of about 0.6 CNY per KWH,

**Table 3 DNSPSO test results.**

| Data set | Resident amount | Total profit (CNY) | | Comparison result (CNY) |
|---|---|---|---|---|
| | | PSO | GA | |
| R101 | 25 | 204.80 | 198.27 | +6.53 |
| R102 | | 87.56 | 85.28 | +2.28 |
| C101 | 50 | 323.65 | 313.03 | +10.62 |
| C105 | | 266.32 | 263.28 | +3.04 |
| RC103 | 100 | 564.84 | 557.31 | +7.53 |
| RC207 | | 551.17 | 541.95 | +9.22 |

then set unit transportation cost is 0.2 CNY/Km, and the carbon emission reduction subsidy coefficient is temporarily set to zero. The average value is recorded as the optimal results. The results are shown in Table 3:

The comparison of examples used in our calculation are shown in Fig. 3. The line in the box represents the median fitness value calculated by PSO for each example, and the blue represents that of GA. It is evident that PSO produces continuous consistent results in small, medium or large-scale cases, while the GA's results are frequently inconsistent. In addition, the median fitness value calculated by GA is lower than that calculated by PSO in each example.

Such comparison may be related to the coding methods of the solutions and the characteristics of the two algorithms. Firstly, from the perspective of the coding method, when using PSO to solve the recycling site location problem, only three digits are needed, and the calculation results of Matlab are automatically accurate to four decimal places. GA coding generally adopts binary coding, and uses a long array composed of only 0 and 1 to express the solution of the site location problem (*Kratica et al., 2001*), which is much more complex than PSO on the premise of ensuring the same accuracy. Secondly, from the perspective of algorithm characteristics, PSO based on swarm intelligence strategy seeks the optimal solution by simulating the foraging of birds within a certain range (*Chen & Tan, 2018*), which has certain similarity with the site location problem in the kernel. However, when finding the optimal solution, GA based on evolution strategy has lots of abstract operations such as chromosome crossover and mutation for site location problem (*Alp, Erkut & Drezner, 2003*; *Rybičková, Mocková & Teichmann, 2019*), which may affect the convergence of the algorithm. Figure 4 shows the convergence of the two algorithms after we run the RC207 example and get the value around average fitness value. In conclusion, we reasonably believe that PSO is effective and robust in solving the proposed carton recycling site location model.

## Model validity analysis

The most important feature of this model is that the number of residents it serves depends on the unit carton recycling price. When we validate the model, we compare it to a model that serves all residents, which implies the unit carton recycling price is set as the

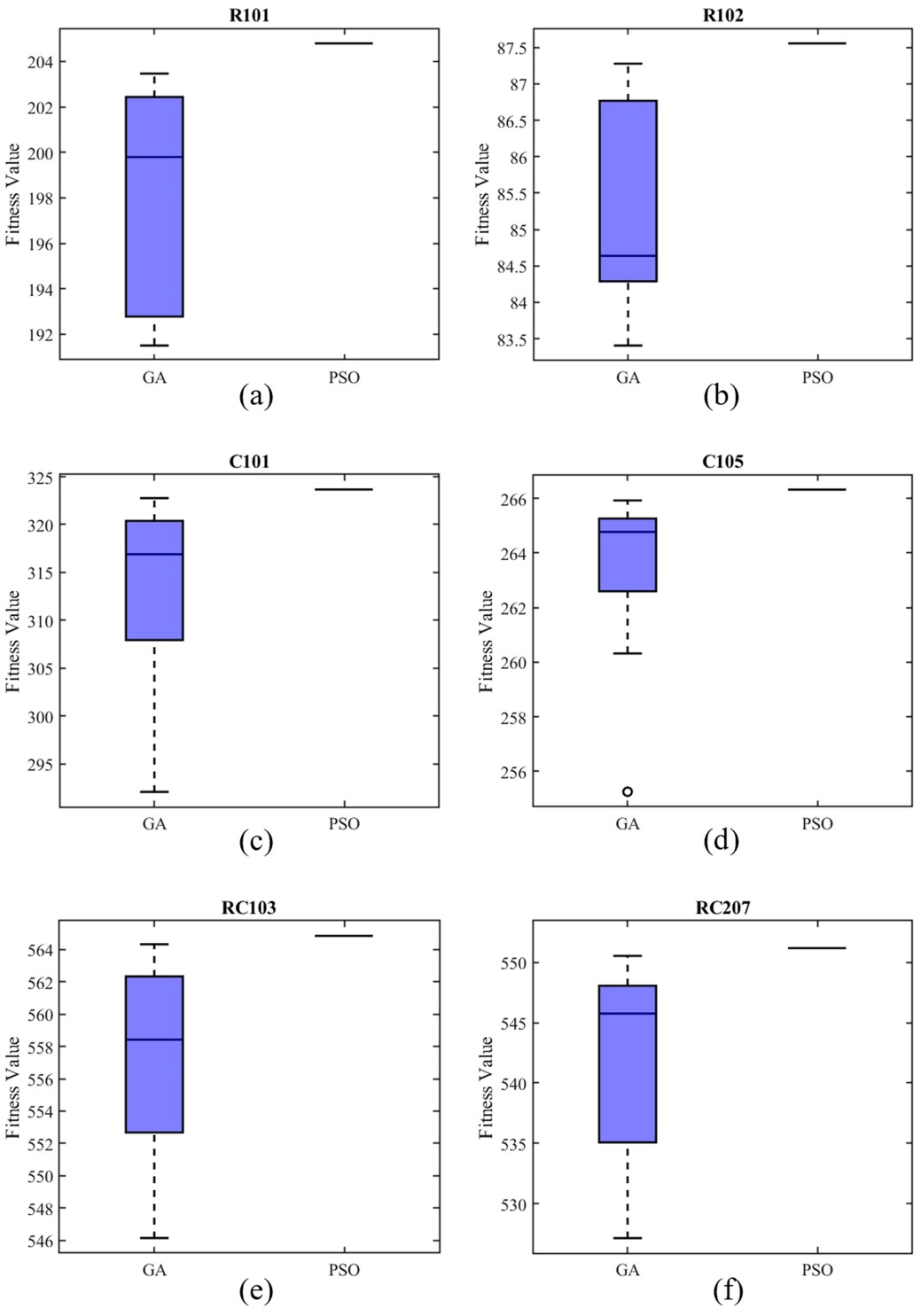

**Figure 3 Results of comparison between PSO and GA.**

highest to meet all residents expectations. We assume that the unit transportation cost is 0.2 CNY/Km, and the carbon emission reduction subsidy coefficient is still temporarily set to zero.

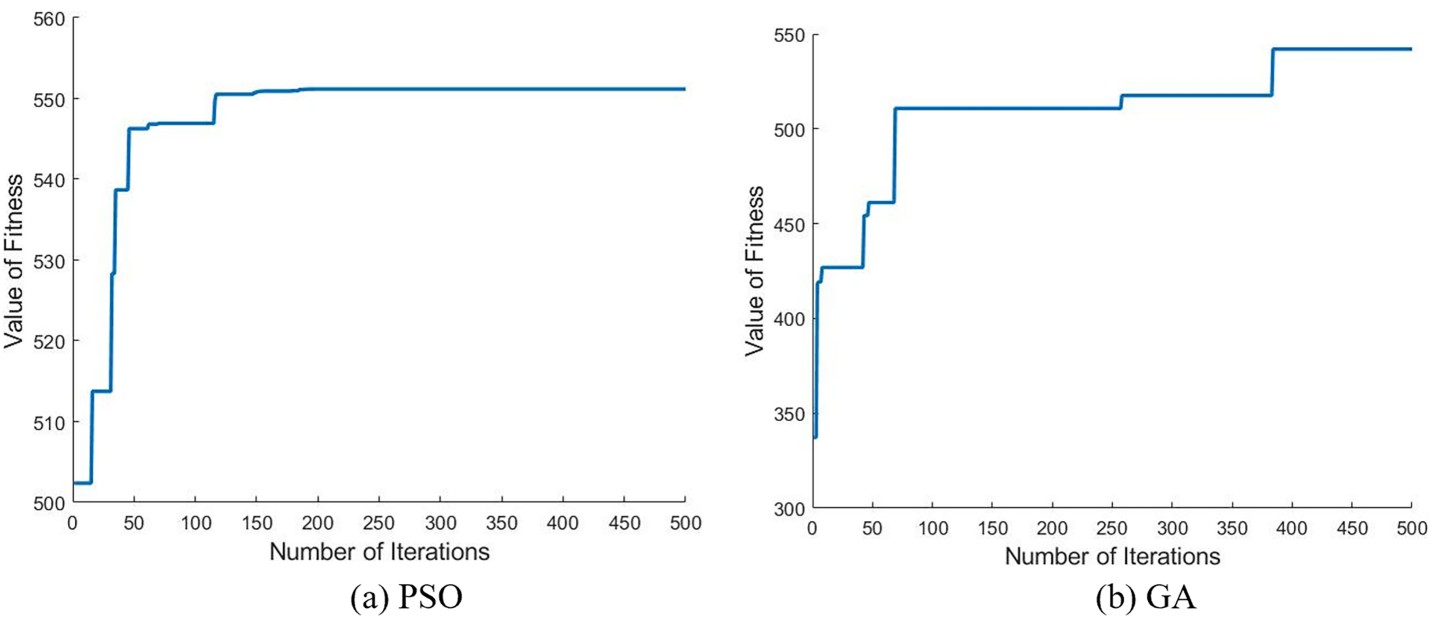

**Figure 4 Comparison of algorithm convergence when obtaining values around the average (using RC207 example).**

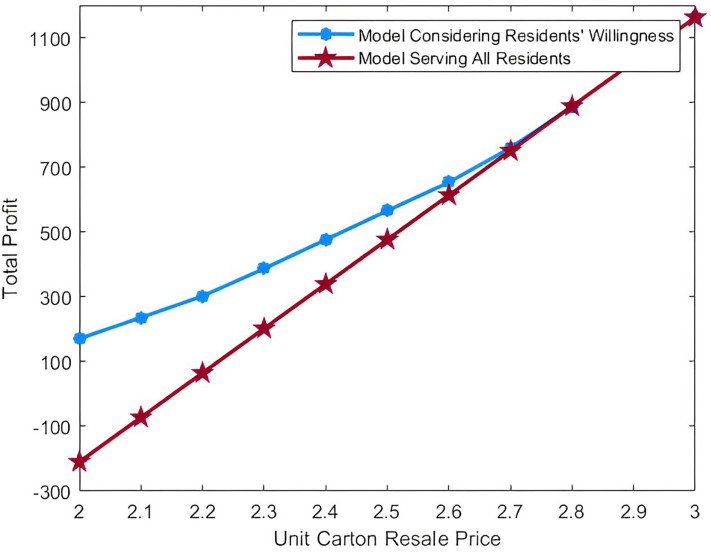

**Figure 5 Comparison of total profit in two models.**

Figure 5 shows the comparison of total profit as the unit carton resale price rises in two models. It is not difficult to see that when the unit carton resale price is low, the model that considers residents' willingness to participate in recycling can get more total profits. This is because when the income from reselling cartons to the paper mill is not sufficient, serving residents with a high price will lead to a loss of the enterprise. However, when the unit carton resale price rises, the gap between the total profits calculated by the two models will

narrow. That is to say, when the income from reselling cartons is sufficient, the enterprise can still get profits by serving the residents with high price. This comparison can prove the validity of our proposed model.

## Sensitivity analysis

After verifying the effectiveness of PSO and the validity of our proposed model, we will complete the analysis and discussion of the proposed carton recycling site location model, that is, sensitivity analysis. Among them, the large-scale calculation example RC207 will be used as the sensitivity analysis, specific data of RC207 example are in the Supplemental Material. Since the model is built from the perspective of the recycling enterprise, and the ultimate goal of recycling enterprises operation is to maximize their total profit, which involves the important decision of the unit carton recycling price. Therefore, we mainly discuss the influence of the variable parameters in the model on the unit carton recycling price offered by the enterprise and the final total profit.

First, we explore the influence of the unit carton resale price and the unit price of transportation per kilometer on the unit carton recycling price and the final total profit of the enterprise, without considering the carbon emission reduction subsidy briefly. According to the actual unit carton recycling price in China, it generally fluctuates between 2 and 3 CNY/Kg, so the unit carton recycling price is set to fluctuate within the range. In addition, we consider a 50% fluctuation in the unit transportation cost. The specific calculation results are shown in Fig. 6. It can be observed from the total profit comparison that the total profit rises steadily with the rise of the unit carton resale price. Figure 6A shows several smooth curves with even spacing. The scenario with lower unit transportation price per kilometer can obtain higher total profit, and with the rise in unit carton resale price, the difference in total profit will gradually increase. In the unit carton recycling price comparison (Fig. 6B), with the rise of the unit carton resale price, price curves show plateaus and sharp increases, and overlap (but not cross) at many nodes. By observing a single curve, it can be found that each curve has several plateaus, that is to say, with the rise in unit carton resale price, the unit carton recycling price may remain stable for a period of time. As the unit carton resale price continues to rise, the unit price of recycling may jump by leaps.

The plateaus in the curve of the unit carton recycling price comparison reflect the precise calculation of the proposed model and algorithm for the maximum benefits of the enterprise. To specifically explain this point of view, through the observation of the example data, we can find that the psychologically expected value of unit carton recycling price of some residents only accounts for a small portion, and the marginal benefit of the enterprise to rise the unit price to serve these residents may be less than the transportation cost, which is the reason for the plateaus in the curve. With the rise of the unit carton resale price, when it reaches a certain value, the curve may enter a new plateau. In addition, due to the large number of residents in the RC207 example, the total profit curve remains steadily growing even though there are many plateaus and sharp increases in the unit carton recycling price curve. That is to say, although the site selection and pricing strategy

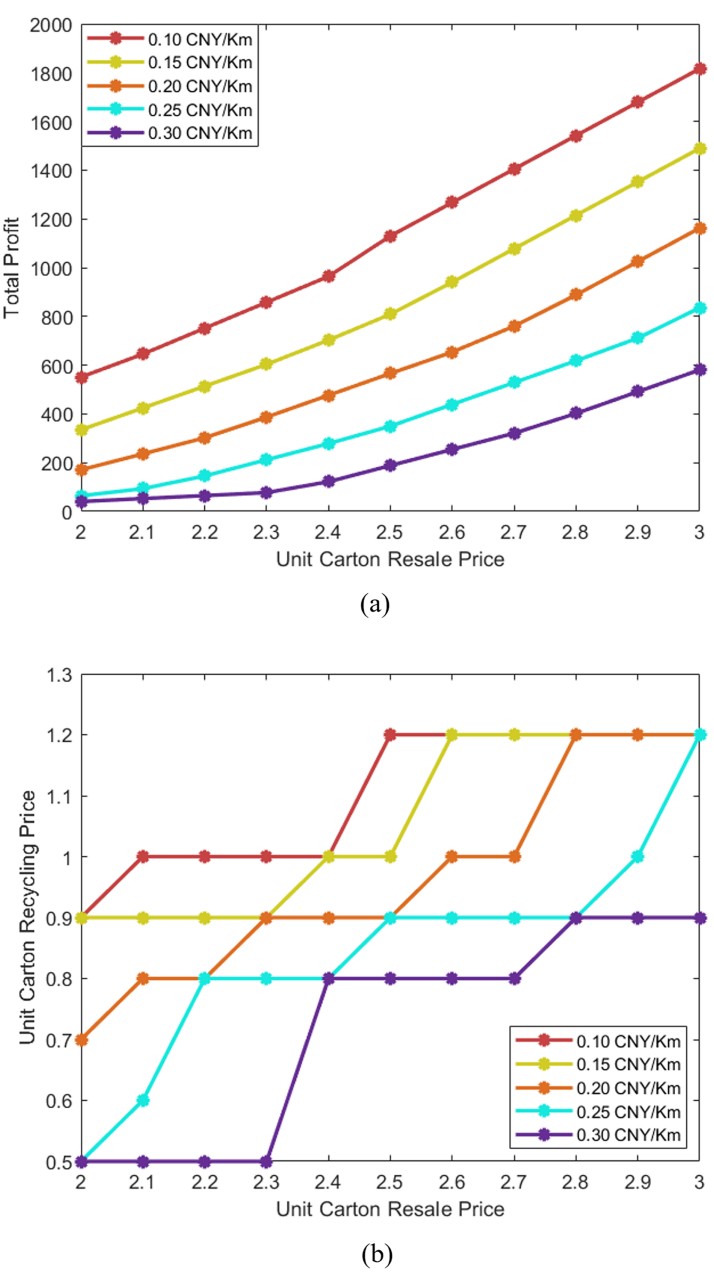

**Figure 6 Comparison of total profit and unit carton recycling price.**

of an enterprise are very sensitive to the influence of other factors, the huge number of services determines that its total profit will not be greatly affected.

We choose a scenario where the unit transportation price per kilometer is 0.2 CNY/Km, and discuss the influence of the carbon emission reduction subsidy coefficient on the overall profit and pricing strategy of the enterprise. See Fig. 7 for specific calculation results. The comparison of total profit and unit carton recycling price are similar to Fig. 6. The results show that under the same unit carton resale price, a higher carbon emission

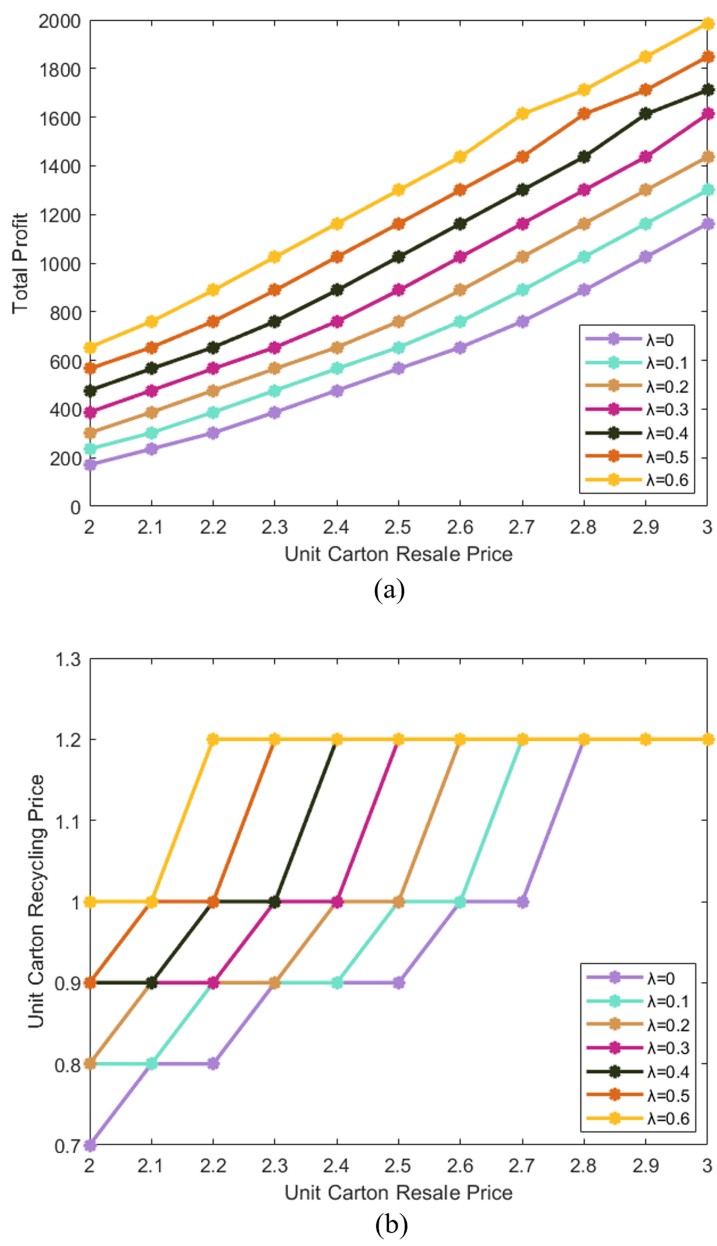

**Figure 7 Comparison of total profit and unit carton recycling price (considering subsidies).**

reduction subsidy coefficient may encourage the enterprise to raise the unit recycling price and collect more cartons, but this is not necessarily the case (the curves may overlap).

Since the sensitivity analysis considers all scenarios with varying parameter combinations, we found that even the PSO algorithm, which is highly stable in the validity test, will repeatedly calculate two results under some scenarios with parameter combinations. See Fig. 8 for details of the two results (unit carton resale price is 2.3 CNY/Kg, transportation cost is 0.2 CNY/Km, carbon emission reduction subsidy coefficient is 0.3).

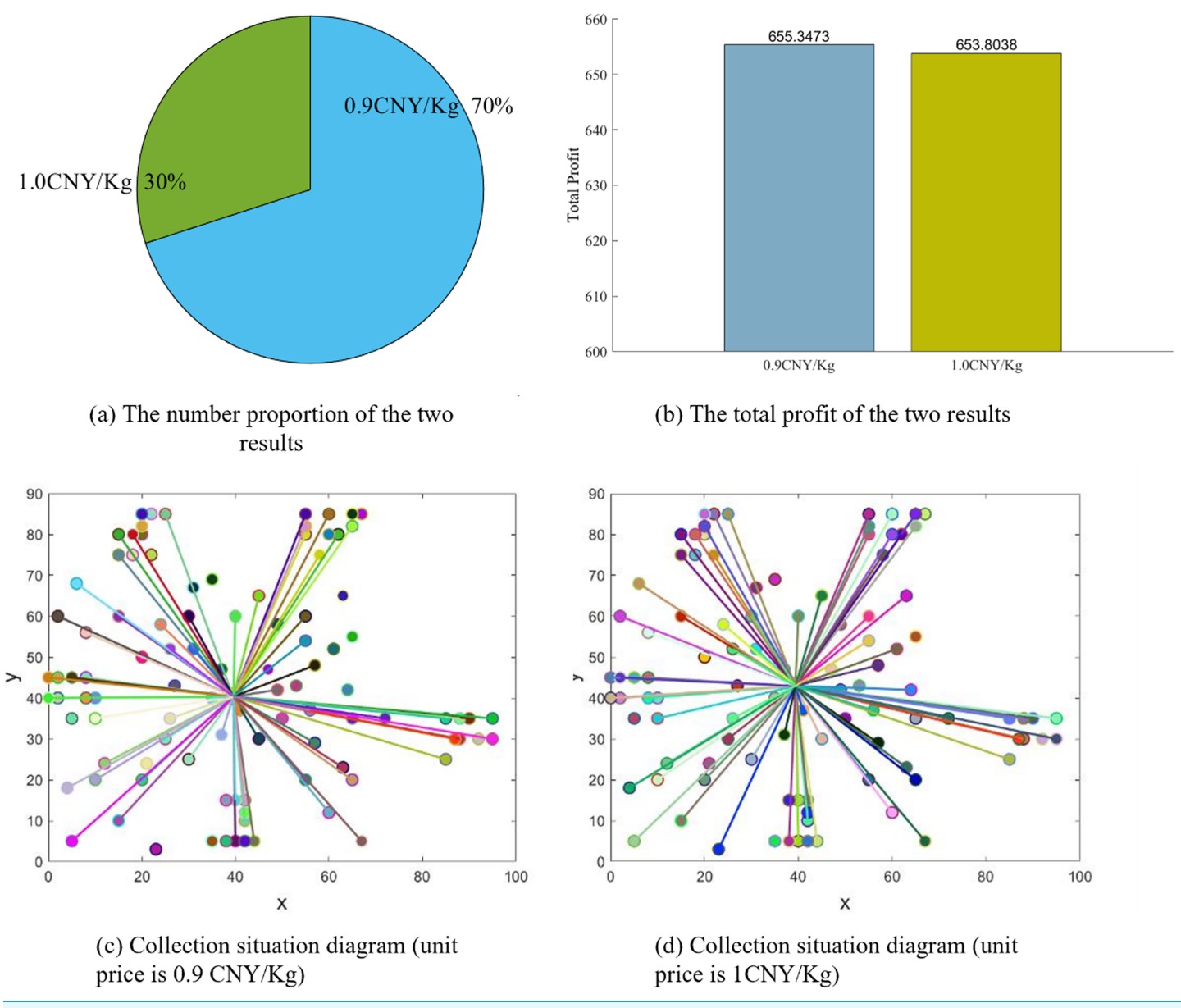

(a) The number proportion of the two results

(b) The total profit of the two results

(c) Collection situation diagram (unit price is 0.9 CNY/Kg)

(d) Collection situation diagram (unit price is 1CNY/Kg)

**Figure 8  PSO calculation results.**                               

These results have some implications for our proposed model and algorithm. At the model level, the two results show that although enterprises set different carton unit recycling price, the final total profits are almost identical, which confirms a point: if the number of residents is large enough, small fluctuations in unit carton recycling price considering the psychological expectations of a small number of residents will not affect the enterprise's overall total profit. At the algorithm level, although the total profit of the two results are similar, the solution with low profit may still be suboptimal. In this case, the solution with the higher profit should be chosen. this also tells us that the PSO algorithm needs to be strengthened.

## Discussion

From the sensitivity analysis, we can know that a higher unit carton resale price and a lower unit transportation price can promote the enterprise to raise the unit carton recycling price and collect more cartons, thus achieving the two goals of contributing to carbon emission reduction and profit maximization. In addition, we came to several important conclusions:

1. The carton pricing strategy is sensitive to cost variables, and higher carbon emission reduction subsidy may encourage the enterprise to raise the unit recycling price and collect more cartons.

2. Although the enterprise's pricing strategy is affected by other factors, the total profit remains stable in the end because the number of residents is large enough. The different pricing strategies made by considering the psychological expectations of a small number of residents towards the carton recycling unit price will not affect the enterprise's overall total profit.

3. It can be concluded from the two points above that the overall decision of an enterprise is influenced by many aspects. Therefore, it has important reference significance to set location selection and pricing as decision variables simultaneously. Compared with first pricing and then site selection or first site selection and then pricing, our approach in theory has more comprehensive consideration and can help to find better solutions.

4. The deviation of the PSO algorithm in the sensitivity analysis on the one hand confirms the second conclusion above, and on the other hand shows that the classical PSO algorithm adopted in this paper still has instability in large-scale site selection research. The latest improved PSO algorithm can be used or combined with the algorithm based on evolutionary strategy in future site selection research.

5. While we utilized a calculation example in the sensitivity analysis, its application to real-world data in the future can further unlock its practical value.

Overall, this model offers valuable insights for recycling enterprises. Operating costs for such enterprises comprise multiple components, which at times can be mutually challenging. Our study integrates site selection and carton pricing strategies to encourage enterprises to approach cost-related issues from a comprehensive standpoint. Moreover, the matter of carton pricing and subsidies involves a complex interplay between enterprises, government entities, and residents. Although this paper did not delve deeply into the intricacies of this game, relying instead on simplified assumptions, future research can explore the dynamics of this game within reverse logistics in greater detail. By effectively combining this aspect with logistics optimization, a more comprehensive theoretical framework can be established.

## CONCLUSION

Waste cartons recycling is a huge benefit to the economy and the environment. The development of the Internet has given birth to new carton recycling mode. Considering the characteristics of this new carton recycling mode and the in-applicability of the previous location research models for it, we reconstruct the carton recycling site location model from the perspective of recycling enterprises. In the model, the unit carton recycling price

and site selection are both decision variables, residents' willingness to recycle considering their psychological expectations regarding the unit carton recycling price and the subsidy income of carbon emission reduction are fully considered. The following are the specific conclusions of our study.

## Main conclusions

Firstly, we have involved the application of PSO algorithm to the solution of the model. In the algorithm validity test, the performance of PSO is more stable than that of GA, and better solutions can be found. Secondly, we validate the model by comparing it to a model that serves all residents, which shows the significant increase in total profit when setting location selection and pricing as decision variables simultaneously. Finally, we did a sensitivity analysis which shows that although the pricing strategy of an enterprise is sensitive, the total profit can remain stable rather than change dramatically after undergoing a fluctuation in pricing strategy. This is because the number of residents is large enough.

## Theoretical significance

Firstly, our research innovatively combines location selection and carton pricing decisions together for collaborative optimization, which can avoid the sub-optimal solution when the two decisions are optimized separately. Secondly, the model proposed by us has high universality and can incorporate other decisions such as inventory optimization or route optimization for more systematic and comprehensive optimization in the future.

## Management significance

The carton recycling site selection model proposed in our study fills the gap in the development of reverse logistics in the new waste recycling mode and lays a good foundation for the exploration of site selection in the field of waste recycling. At the same time it provides a great reference value for decision-making in recycling enterprises: although enterprises have very mature research and use experience on site location selection and pricing decision-making in reality, these two decision-making tasks sometimes belong to two different departments, which may lead to the generation of sub-optimal solutions, and thus increase the operating costs of enterprises. In our proposed model, the idea of collaborative optimization between the two can provide new thinking for the operation of enterprises.

## Limitations and future direction

Future studies can make some improvements in these ways: leaving the number of sites unfixed to solve a multi-site location problem considering site opening costs, combining path optimization and site location together to build a site location route optimization model.

## ACKNOWLEDGEMENTS

Throughout the writing of this article I have received a great deal of support and assistance. Finally, thanks to the support of all my partners and family members, I was able to finish this article.

### Funding

This work was supported by the Fundamental Research Funds for the Natural Science Foundation Project of Chongqing (cstc2019jcyj-msxmX0100), the Fundamental Research Funds for the Central Universities (No. 2020CDJSK03XK13) and the National Social Science Fund of China (No. 21BGL127). The funders had no role in study design, data collection and analysis, decision to publish, or preparation of the manuscript.

### Grant Disclosures

The following grant information was disclosed by the authors:
Natural Science Foundation Project of Chongqing: cstc2019jcyj-msxmX0100.
Central Universities: 2020CDJSK03XK13.
National Social Science Fund of China: 21BGL127.

### Competing Interests

The authors declare that they have no competing interests.

### Author Contributions

- Yulong Xi conceived and designed the experiments, performed the experiments, analyzed the data, performed the computation work, prepared figures and/or tables, and approved the final draft.
- Fengming Tao conceived and designed the experiments, authored or reviewed drafts of the article, and approved the final draft.
- Schanelle Brooks conceived and designed the experiments, authored or reviewed drafts of the article, and approved the final draft.

### Data Availability

 Code and raw data are available in the Supplemental Files.

### Supplemental Information

Supplemental information for this article can be found online at http://dx.doi.org/10.7717/peerj-cs.1519#supplemental-information.

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
