# Peer review of "Optimization of carton recycling site selection using particle swarm optimization algorithm considering residents’ recycling willingness"

_PeerJ Computer Science, doi:10.7717/peerj-cs.1519_

## Round 0.1 · original submission · Major Revisions

More detailed explanations are needed for a better understanding of PSO and GA performance. I suggest revisions in line with the recommendations of the referees.

Reviewer 1 ·

Basic reporting

no comment

Experimental design

no comment

Validity of the findings

no comment

Additional comments

In the study, particle swarm optimization is proposed for the optimization of the site selection problem. The organization of the article, the language of writing and the results are successful. However, the novelty of the article is limited. The discussion section is insufficient. The charts are low resolution and the texts are very difficult to read. Giving the results in tabular form can also be evaluated by the authors. For these reasons, I would like to regret that the acceptance of this article is not appropriate.
Kindly regards

·

Basic reporting

The authors focus on a carton recycle problem in which a recycling company picks up carton from households when the household is willing to sell the carton at a price provided by the company. An optimization problem is formulated and then solved using the PSO metaheuristic optimization algorithm. Then the results are compared against a genetic algorithm.

The problem is interesting and the research is sound but the paper needs some minor changes.

In lines, 97-98, it is not clear why are you comparing a multi-criteria approach such as AHP with metaheuristic optimization algorithms, not all which are multi-objective algorithms.

Line 106. It is not clear when you say that PSO" is an ideal method to solve the carton recycling site selection problem". Is the problem n-hard? What are the reasons behind using a metaheuristic algorithm for this problem?

Experimental design

The assumption in lines 159-160 is already explained in lines 138-141.

The concept of homogeneous cartons should be explained in more detail.

Model assumption (6) in line 173 is not clear.

Does wt replace w in equation (12)?

What is a Solomon benchmark?

What are the crossover and mutation parameters of the GA? What is the population size? What is the termination criteria? What is the selection method that you used (roulette selection, tournament, etc.)? What is the number of generations?

The graphs in Figure 3 show fitness values for different running times. However, the explanation in lines 315-317 indicates that what is represented in that figure is the average fitness for each example.

In your paper, does the fitness function equals the objective function?

Validity of the findings

The authors should report the mean, worst, and best value of the fitness functions for both the PSO and GA. I suggest to use box and whisker plots to compare the PSO and GA results.

Additional comments

Figure 1 should be improved.

Please use the normal flowchart symbols in Figure 2.

In line 483 when you write dissertation, I assume you mean paper.

---

## Round 0.2 · Minor Revisions

The article needs minor revision in line with referee 2's requests.

Reviewer 1 ·

Basic reporting

no comment

Experimental design

no comment

Validity of the findings

no comment

Additional comments

The authors have remarkably emphasized the contribution of the study to the literature. Images are more understandable. After the revisions the manuscript is more successful. As a result, the authors made the necessary corrections. It is appropriate to publish the article.

·

Basic reporting

The authors have addressed all my comments, but the paper still needs some minor corrections to make it more precise and clear. For example, in lines 170-171, it is not clear when authors say that they "care about the weight of the cartons and do not care about their capacity, soundness and other characteristics". What is the capacity of a carton? What is the soundness of carton?

In line 173, what do you mean with "recycling price is unified"?

What do you mean with "psychological expectation"? (See for example line 96)

Experimental design

The authors have addressed all my comments in this regard.

Validity of the findings

The authors have addressed all my comments in this regard.

Additional comments

None.

---

## Round 0.3 · accepted · Accept

I confirm that the revisions have addressed all the concerns raised during the review process and have enhanced the article's contribution to the field.